# The Influence of Non-Uniformities on the Mechanical Behavior of Hemp-Reinforced Composite Materials with a Dammar Matrix

**DOI:** 10.3390/ma12081232

**Published:** 2019-04-15

**Authors:** Dumitru Bolcu, Marius Marinel Stănescu

**Affiliations:** 1Department of Mechanics, University of Craiova, 165 Calea Bucureşti, 200620 Craiova, Romania; dbolcu@yahoo.com; 2Department of Applied Mathematics, University of Craiova, 13 A.I. Cuza, 200396 Craiova, Romania

**Keywords:** composite materials, hybrid resin, natural reinforcement, non-uniformities, mechanical behavior

## Abstract

As a result of manufacture, composite materials can appear to have variations to their properties due to the existence of structural changes. In this paper, we studied the influence of material irregularity on the mechanical behavior of two categories of bars for which we have used hemp fabric as a reinforcing material. The common matrix is a hybrid resin based on Dammar and epoxy resin. We molded two types of bars within each of the previously mentioned categories. The first type, also called “ideal bar”, was made of layers in which the volume proportion and the orientation of the reinforcing material was the same in each section. The ideal bar does not show variations of mechanical properties along it. The second type of bar was molded to have one or two layers where, between certain sections, the reinforcing material was interrupted in several segments. We have determined some mechanical properties, the characteristic curves (strain-stress), the tensile strength, and elongation at break for all the sample sets on trial. Moreover, we have studied the influence of the non-uniformities on the mechanical behavior of the composites by entering certain quality factors that have been calculated after experimental determinations.

## 1. Introduction

Fiber-reinforced composites are widely used for their advantages over non-reinforced materials. Their applications include construction (see [1]), aerospace (see [2]), medicine (see [3]), and dentistry (see [4]) fields.

The mechanical behavior of composite materials is influenced by environmental factors (humidity, temperature, radiation, chemical agents, see [5,6,7,8]), and the mechanical stresses to which they are subjected (the stress type, the stress variation in time, the loading speed, the stress direction, the stress duration, see for example [9,10]).

An important aspect concerning the mechanical properties of composite materials is given by the non-uniformities that appear in the technological process of fabrication. In the case of fiber-reinforced composites the uneven distribution of the matrix-fibers represents the main factor influencing their mechanical behavior. The resin transfer, the structural reactions, and the interface effects are phenomena that can be taken into account when considering the non-uniformity degree (see [10,11]).

Due to the unevenness of the fiber distribution, experimental discrepancies often occur with respect to the physical properties of the studied samples. References [11,12,13] investigate the influences of non-uniformities on composite material behavior; in these works, the elastic constants and density are assumed to be functions of fiber volume proportion, while the distribution of the fiber volume proportion is given by a polynomial function. Reference [14] studies non-uniformities occurring in composite bars, reinforced with a glass fiber fabric, and introduces a coefficient that estimates the non-uniformities of the composite bars with two areas with different volume proportions of the reinforcement. References [15,16] study the non-uniformities occurring in composite bars reinforced by carbon fabric, and by carbon and Kevlar fabric.

The non-uniformities and discontinuities can be used with a view to obtaining composite materials with controlled properties. The traditional laminated composites, reinforced by fibers, are made of multiple laminae in which the fiber-reinforcing direction is constant. Property control is exercised only by modifying the layer sequence in the laminate. Reference [17] checks on the properties of the composite materials by using so-called “mosaic” composites. Each layer in these composites is made of several pieces, each piece with its own type of fiber orientation, length, and distribution. References [18,19,20] showed that using this kind of element in regulated assemblies, which are intertwined, led to obtaining composites with a tensile strength up to 90% compared to the composites with continuous reinforcement, although the mere juxtaposition of such layers may bring about a decrease in the tensile strength by 50%. They made composite materials that had improved damage tolerance because there is a mechanism of slowing crack propagation fissure speed by decreasing the cohesion between the adjacent elements. A theory that anticipates crack initiation and propagation is studied in [21] and its effects on fiber-reinforced composites polymerized with light materials are investigated in [22].

The use of hemp fibers as reinforcement for composite materials has risen in the past few years as a result of an increasing demand for the development of biodegradable and recyclable materials. Much attention has been paid in recent decades to composite materials, the components of which, be it matrix or reinforcement, come from nature. The advantages of using green composites lie in both the fibers and the bio-resins being abundantly made by nature, consequently having a low manufacture cost compared to synthetic composites. Moreover, they have relatively good mechanical properties. Hemp fibers can be found in the plant stem, which makes them tough and rigid, an essential requirement for the reinforcement of composite materials. The mechanical properties of hemp fibers are close to those of the glass fibers (see [23,24,25,26,27]). Nevertheless, the biggest disadvantage is the variability of these properties depending on the weather conditions, the harvest area, or other natural factors. In [28], the author explains why the hemp fibers from plants harvested at the beginning of flowering had a stiffness and a tensile strength greater than the fibers of the hemp plant seeds harvested at maturity. In [29] it is shown that the fibers of the middle section of the stock had higher mechanical properties than the fibers of the upper and lower sections of the stem. Studies on the mechanical properties of the hemp fibers are described in detail in [30]. Improvement of the mechanical behavior of hemp fibers, as well as different types of treatments, is shown in [31,32].

Most composite materials studied so far have focused on natural fibers as reinforcing materials in combination with thermoplastic matrices (polypropylene, polyethylene, and vinyl polychloride), or thermo-rigid matrices (phenolic, epoxy, and polyester resins) [33]. The composites made of hemp fibers with thermoplastic, heat resistant, and biodegradable matrices have shown good mechanical properties. In addition, the surface treatments applied to the hemp fibers, with the purpose of improving the bond on the interface between the fibers and the matrix, have led to considerable improvements in mechanical properties. Some mechanical properties which characterize tensile mechanical behavior, torsion, and bending the polypropylene matrix composite material and reinforced with hemp are shown in [34,35,36].

The synthetic resins have the disadvantage of a processing limit due to high viscosity when melted, a phenomenon that occurs during injection molding. Thus, the final product is hard to recycle. This disadvantage can be eliminated by using thermo-rigid biological matrices, based on vegetable oil resins, since the latter are biodegradable and there is no need for a polymerization process [37,38,39]. The bio-resins are resins derived from a biological source, and in general they are biodegradable or compostable; therefore, hypothetically, they can be decomposed after use. The natural resins may be fossil (amber), vegetable (Sandarac, Copal, Dammar), or animal (Shellac). Natural resins are insoluble in water but are slightly soluble in oil, alcohol, and, partially, petrol. They form, together with certain organic solvents, solutions used as covering varnishes. Turpentine, rosin, and mastic are products resulting from coniferous resin distillation. An analysis of the chemical composition and the properties of these resins is made in [40] and the applications are presented in [41].

The analysis of these resins focused more on their chemical composition and chemical properties, presented in [42,43], and less on their mechanical properties. For Dammar, which is a gum resin obtained from trees of the family Dipterocarpaceae from India and East Asia, detailed studies on the structure and chemical composition are made by [44].

Reference [45] presents a new silicon and Dammar-modified binder that reduces the use of synthetic binder and that has improved and more ecological properties. The optimal composition for this binder was determined, ensuring the best properties for impact stress, hardness, traction, and adherence. Reference [46] studies how the Dammar addition contributed to improving the rigidity, elasticity modulus, and hardness of a modified silicone.

There are quite a few analyses of the mechanical behavior of natural resins. Reference [47] investigates the mechanical features (tensile strength, percentage elongation and Young’s modulus), the water vapor transmission characteristics, and the characteristics of humidity absorption in Dammar films that contain and that do not contain softening agents. The reaction to the compression stress of the oil palm trunk treated with various amounts of Dammar resin was studied in [48]. Also, there are few studies of composite materials with both the matrix and the reinforcement of natural materials. References [49,50] examine the mechanical behavior of some composite materials with a matrix of Dammar-based resin and the reinforcement of cotton, flax, silk, and hemp fabric.

Since natural lakes can form thick resin (see for example references [51,52]), we can conclude that the bio-resins were studied before actual hybrid resins. A hybrid resin is a combination of several constituents, of which at least one is organic and synthetic. It should be noted that most attempts to obtain such resins have been varnishes (see [51,52,53]). Hybrid resins are alternative environment-friendly compared with synthetic resins. Some practical and useful notes about this problem can be found in the papers [54,55]. Larger investigations about hybridization (in fibers and/or matrices) are available in [56,57].

## 2. Materials and Methods

### 2.1. Making the Samples

The first step consisted of casting hybrid resin plates where we used a Dammar volume proportion of 60%. The difference up to 100% consisted of epoxy resin of Resoltech 1050 type together with its associated hardener Resoltech 1055. The casting temperature was 21–23 ${}^{\circ }$C. To realize the Resoltech 1050/Resoltech 1055 combination we respected the manufacturer’s instruction. We used a mixture ratio of 7/3 after a given volume. We mixed the epoxy resin obtained with Dammar resin. All samples based on hybrid resin were cut after 10 days.

The synthetic component (the epoxy resin and the reinforcing material) was necessary to generate quick points of activation of the polymerization process. The thermo-mechanical properties of Resoltech 1050 epoxy resin, together with its associated hardener Resoltech 1055, are given by the producer (see [58]).

Figure 1 shows a hybrid resin sample.

The second step consisted of making Dammar-based composite materials by reinforcing them with two types of hemp fabric. The first type of fabric, symbolized by “A” is shown in Figure 2a, the fabric with the characteristic mass of 330 g/cm^3^. For this type of fabric, on a surface of 10 cm × 10 cm, we had 50 strands oriented in the direction of the tensile test (longitudinal direction) and 33 strands in transversal direction. The second type of fabric, symbolized by “B”, is presented in Figure 2b, the characteristic mass of this fabric being 350 g/cm^3^. In this type of fabric, on a surface of 10 cm × 10 cm, we had 62 strands oriented in the direction of the tensile test (longitudinal direction) and 28 strands in transversal direction. The plates obtained had five layers of fabric. In the plates considered as ideal, all five layers are continuous, without interruption. To identify the studied composite materials, we use symbols made of one letter and two digits. The letter may be “A” or “B”, depending on the type of fabric used. The first digit is 0, 1, or 2 and represents the number of interrupted layers. For instance, A24 stands for composite material with a reinforcement of the type “A” fabric, with two interrupted layers and the interruption length of the layers of four centimeters.

The hemp fiber properties are (see [37,50,59]): 1.4–1.5 g/cm^3^ density, 30–60 GPa modulus of elasticity, 310–750 MPa breaking load, 2–4% elongation at break.

We produced sets of 10 samples that were 250 mm long and 25 mm wide. These sizes are employed in the tensile tests performed according to ASTM D3039 (see [60]).

In Table 1 we present details about manufactured samples.

### 2.2. Theoretical Considerations

We have a straight bar, with the length *l*, square section, width *b* and thickness *h*, made of *n* layers of composite materials of constant thickness. A reference system is attached to the bar, with the axis Ox along the bar. The bar is submitted to a tensile test along the Ox axis. We accept that a plane and normal section on the Ox axis before the stress stays plane and normal on this axis throughout the test.

In this hypothesis, the characteristic deformation along the bar depends only on the abscissa *x*, thus $\varepsilon =\varepsilon \left(x\right)$.

The normal tension in layer *k* in the section of the abscissa *x* is (see [14]):(1)$$\sigma_{k}\left(x\right)=E_{k}\left(x\right)\cdot \varepsilon \left(x\right),$$
where $E_{k}\left(x\right)$ is the elasticity modulus of the layer *k* in the section of the abscissa *x*.

The tension resultant in the section of the abscissa *x* is the traction force to which the bar is submitted:(2)$$F=\underset{\left(S\right)}{\;\int \int \;}\sigma \left(x\right)dS=b\sum_{k=1}^{n}h_{k}E_{k}\left(x\right)\cdot \varepsilon \left(x\right),$$
where $h_{k}$ is the thickness of the layer *k*.

The average modulus of elasticity of the bar can be determined by the ratio:(3)$$E_{med}=\frac{F\cdot l}{\Delta l\cdot A}=\frac{l}{h\int_{0}^{l}\frac{1}{\sum_{k=1}^{n}h_{k}E_{k}\left(x\right)}dx}.$$

If in a layer *k*, in the section of the abscissa *x*, its breaking load reaches $\sigma_{r}^{\left(k\right)}\left(x\right)$, then it breaks and a first bar damage occurs. After the break of the first layer, the efforts are taken over by the other layers, causing a tension change in the layers. Thus, there are two scenarios for the break mechanism. The first presupposes that the breaking load is reached in other layers too, hence their concatenation and the bar break without any increase in the stress force. The second presupposes that after the redistribution of the tensions in the layers the breaking load is not reached in any of the remaining layers. In this case, the stress force may increase, leading to tension increase until the tensile strength is reached in one of the remaining layers, repeating the break phenomenon. The breaking load of the bar, marked by $\sigma_{r}$, is considered to be the average tension in the bar section at the moment when the concatenated break and the bar destruction occur. Since Hooke’s law stipulates proportionality between tensions and characteristic deformations, the highest tensions appear in the sections where the characteristic deformations are maximal. These arise in the points where the bar section rigidity is minimal. In practice, the break can be considered to occur at the strands in the area where the bar section rigidity is minimal.

We have two bar types. The first type, called ideal, is made of layers that do not show property variations along the bar. In this type of bar, the volume proportion and the reinforcement orientation, for each layer, is considered to be the same in any section of the bar. The second bar type is considered to have layers where, between certain sections, the reinforcement is removed and the resin remains alone.

The second type of bars can be regarded as bars with different non-uniformities occurring because of the defects that appear during the technological process of bar making. We define the following factors characterizing quality:-the elasticity factor
(4)$$f_{E}=\frac{E}{E_{ideal}},$$
where *E* is the elasticity modulus of the analyzed sample and $E_{ideal}$ is the elasticity modulus of the material considered to be ideal, without uniformities;-the resistance factor
(5)$$f_{\sigma }=\frac{\sigma_{r}}{\sigma_{r\;ideal}},$$
where $\sigma_{r}$ is the tensile strength of the analyzed sample material and $\sigma_{r\;ideal}$ is the tensile strength of the material considered as ideal, without non-uniformities;-the uniformity factor
(6)$$f_{u}=\frac{f_{\sigma }}{f_{E}}.$$

These three factors give information on the properties of the analyzed sample material, properties that are compared to those of the reference material, which is considered ideal. The decrease in the elasticity factor shows the decrease in the modulus of elasticity and the decrease of the resistance factor shows the decrease in the tensile strength of the analyzed materials. The decrease in the roughness factor shows the presence of some areas in the material with discontinuities of the mechanical properties.

A particular case is that of the samples with length *l*, made of *n* layers, in which at $\left(n-k\right)$ layers, which are identical, the reinforcement has the same proportion and orientation along the whole length, and at *k* layers, between two sections situated at a distance $l_{0}$, the reinforcement is removed and replaced by resin. In this case, the elasticity factor and the resistance factor are expressed by:(7)$$f_{E}={\left[1-\frac{l_{0}}{l}+\frac{l_{0}}{l}\frac{1}{1-\frac{k}{n}+\frac{k}{n}\frac{E_{r}}{E_{s}}}\right]}^{-1},$$
(8)$$f_{\sigma }=1-\frac{k}{n}+\frac{k}{n}\frac{E_{r}}{E_{s}},$$
where $E_{s}$ is the elasticity modulus of a layer without discontinuities and $E_{r}$ is the elasticity modulus of the resin replacing the removed reinforcement.

## 3. Experimental Determinations

The obtained samples were submitted to a tensile test, which was performed according to the ASTM D3039 provisions (see [60]). We used the testing machine for mechanical trials LLOYD LRX PLUS SERIES (the manufacturer’s data [61]) with the following features:-force range: 5 kN;-travel: 1 to 735 mm;-crosshead speed: 0.1 to 1020 mm/min;-analysis software: NEXYGEN.

Figure 3 presents the installation for a hybrid resin sample tensile test.

Figure 4 shows the characteristic curve of a Dammar-based resin sample. The experimental results for the set of hybrid resin samples are:-tensile strength: 21–22 MPa;-elongation at break: 1.95–2.20%;-modulus of elasticity: 1130–1220 MPa.

The samples made of hemp-reinforced composite materials were also subject to tensile testing.

Figure 5, Figure 6 and Figure 7 show the characteristic curves of the composite materials reinforced by type “A” fabric.

Figure 5 presents the characteristic curve for a sample reinforced with hemp fabric of type “A” without reinforcement-interrupted layers.

Figure 6 presents the characteristic curves for samples reinforced with hemp fabric of type “A” which have an interrupted layer of reinforcement.

Figure 7 presents the characteristic curves for samples reinforced with hemp fabric of type “A” which have two interrupted layers of reinforcement.

For every sample reinforced with type “A” hemp fabric, we give the below (Table 2) the lower and upper values (arithmetical average value and deviation value) for the elasticity modulus, tensile strength, and elongation at break. We have not made a statistical analysis for those values simply because all the outcomes are strictly related to samples in the study. The mechanical properties of hemp fibers come under many influences, so a statistical analysis in this case would have not been relevant.

Figure 8, Figure 9 and Figure 10 present the characteristic curves of the composite materials reinforced by type “B” fabric.

The Figure 8 shows the characteristic curve for a sample reinforced with hemp fabric of type “B” without reinforcement-interrupted layers.

Figure 9 presents the characteristic curves for samples reinforced with hemp fabric of type “B” which have an interrupted layer of reinforcement.

Figure 10 presents the characteristic curves for samples reinforced with hemp fabric of type “B” which have two interrupted layers of reinforcement.

For every sample reinforced with type “B” hemp fabric, we give, in the below (Table 3), the lower and upper values (arithmetical average value and deviation value) for the elasticity modulus, tensile strength, and elongation at break.

## 4. Conclusions

The presence of areas with low mechanical properties in composite materials can lead to alteration in the overall properties of the composite materials. Meanwhile, industrial finished products with variable mechanical properties can be obtained that have a mechanically controlled behavior. Specifically, materials can be obtained that should yield to certain applications, and the breakage should occur in areas particularly chosen from the design.

For all the specimens, which have one or two interrupted layers, both the modulus of elasticity and the breaking resistance are proportional to the modulus of elasticity and the tensile strength of the materials considered as ideal. Thus, the ratio of the average modulus of elasticity of the A00-abbreviated specimens and the average modulus of elasticity of the B00-abbreviated specimens is 0.687. Similar values are obtained for the ratios between the modules of elasticity to the composite materials reinforced with hemp weaving “A” and the modules of elasticity to the composite materials reinforced with hemp weaving “B” (ratios are calculated between the modules of elasticity of the specimens with the same number of interrupted layers and the same length of the discontinued area). The ratio of the average breaking strength of the A00-abbreviated specimens and the average breaking strength of the B00-abbreviated specimens is 0.747. Similar values are obtained for the ratios of the tensile stress for composites reinforced with woven hemp type “A” and the tensile stress for composites reinforced with woven hemp type “B” (ratios are calculated between the modules of elasticity of the specimens with the same number of interrupted layers and the same length of the discontinued area).

It is noted that both the elastic modulus and the tensile stress decreased to increase the number of discontinued layers. Tensile strength decreases to increase the number of interrupted layers regardless of length of the interruption zone. The modules of elasticity for the specimens to the length of the discontinued zone is zero (interrupted layers have the reinforced sectioned in place), and are close to the modules of elasticity of the reference specimens that were considered ideal. For the test pieces with interrupted zone length of 20 mm or 40 mm, the modulus of elasticity decreases to increase the number of discontinued layers. In addition, the modulus of elasticity decreases to increase the length of the interrupted area. A particular trend is found for the elongation at breaking. Both for composite materials reinforced with hemp type “A” and those reinforced with hemp type “B” of the reference specimens (indicated A00 and B00) present the highest breaking elongation. Increasing the number of interrupted layers leads to a decrease of breaking elongation. However, for specimens with the same number of interrupted layers, the breaking elongation increases to increase the length of the discontinued area.

Therefore, the studied composite bars properties depend on:-the properties of the component materials (modulus of elasticity $E_{s}$ and the tensile strength $\sigma_{r}^{\left(s\right)}$, of the reference material, respectively modulus of elasticity $E_{r}$ of the matrix);-number of interrupted layers (the ratio $\frac{k}{n}$ between the number of interrupted layers and the total number of layers);-the interruption length of the layers (the ratio $\frac{l_{0}}{l}$ between the interruption length and the bar length).

We believe that the reference values of the elasticity modulus and tensile strength of the materials considered as ideal are the average values of the limits between which these sizes vary in samples with all their layers without discontinuities (see Table 2 and Table 3). For type-A00 samples the reference value of the elasticity modulus is 4547 MPa and the tensile strength is 56 MPa. For type-B00 samples the reference value of the elasticity modulus is 6617 MPa and the tensile strength is 75 MPa. We notice that both the elasticity modulus and the tensile strength of the composites reinforced by type “B” fabric have significantly higher values than those of the composites reinforced by type “A” fabric. This can be explained by the different number of strands oriented in the direction of the tensile test (50 for type “A” fabric and 62 for type “B” fabric). The fact that the elasticity modulus of the samples in the reference sets is in correlation with the elasticity modules of the fabrics used for reinforcing, and with the number of fibers oriented in the direction of the stress, shows that the longitudinal modulus of elasticity of the composite material is proportional to the modulus of elasticity of the reinforcing material and its volume proportion, just as in classic mixture theory.

Calculating, in a similar way, the reference values of the elasticity modulus and tensile strength for each of the analyzed sample sets, the presented quality factors can be determined.

Table 4 shows these quality factors for the sample sets reinforced by type “A” fabric and in Table 5 for type “B” fabric.

Analyzing Table 4 and Table 5 leads us to the following conclusions:-the elasticity factor decreases if the number of interrupted layers increases; this shows that if the number of layers with interruptions increases, the elasticity modulus of the composite decreases;-the elasticity factor decreases if the layer interruption length increases; this shows that if the length of the area where resin replaces reinforcement increases, then the elasticity modulus of the composite decreases;-the resistance factor decreases if the number of interrupted layers increases; therefore, the tensile strength decreases if the number of interrupted layers increases;-the resistance factor is not influenced by the layer interruption length; this shows that breaking can occur when the fibers in the area where the bar section rigidity minimal break up;-the uniformity factor decreases if the number of interrupted layers increases;-the uniformity factor increases if the layer interruption length increases.

## 5. Discussion

Three factors give information on the properties of the analyzed sample material, properties that are compared to those of the reference material, which is considered to be ideal. Values close to 1 for the three factors point out that the analyzed material has properties that are very close to those of the reference material. A decrease in the values of the three factors indicates the presence of manufacture defects or that the analyzed material is different from the reference material. A value of less than 1 of the resistance factor shows that there are areas in the material where the mechanical properties are inferior to those of the reference material. In the case of composite materials, it indicates that there are areas where either the reinforcement volume proportion is lower than necessary, or the reinforcement orientation does not coincide with that of the reference material. A value less than 1 of the elasticity factor may give information on the dimensions of the area where the mechanical properties are inferior to those of the reference material. The lower the elasticity factor, the larger the faulty area dimensions. If the elasticity factor is 1 and the resistance factor is less than 1, there is a concentrated defect in the analyzed material. The uniformity factor characterizes the uniformity of the analyzed material. Decreased values of this factor point out the existence in the material of certain areas where the mechanical properties differ very much. There might be a case when the resistance factor and the elasticity factor have values less than 1, be very close, or a case in which the uniformity factor is near to the value 1. This shows that the analyzed material is uniform and so different from the reference material that it is actually a different material and not the faulty reference material. In the case of composites, this situation may indicate that the analyzed material has reinforcing material proportions, resin specifically, different from those of the reference material.

## Figures and Tables

**Figure 1 materials-12-01232-f001:**
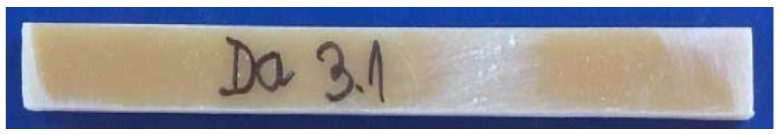
Hybrid resin sample.

**Figure 2 materials-12-01232-f002:**
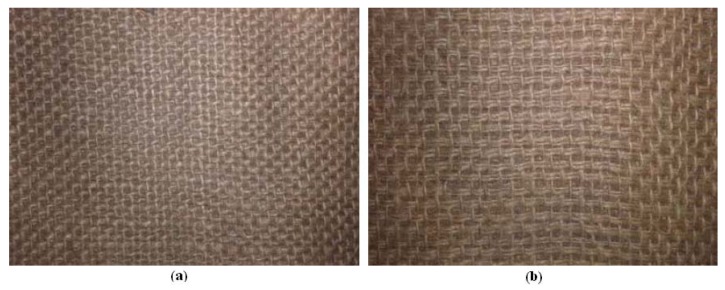
Hemp fabric: (**a**) type “A”; (**b**) type “B”.

**Figure 3 materials-12-01232-f003:**
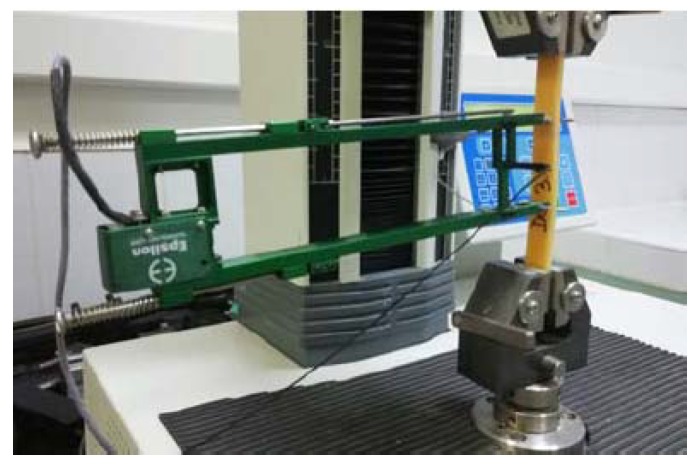
The tensile test assemblage of a sample from the hybrid resin.

**Figure 4 materials-12-01232-f004:**
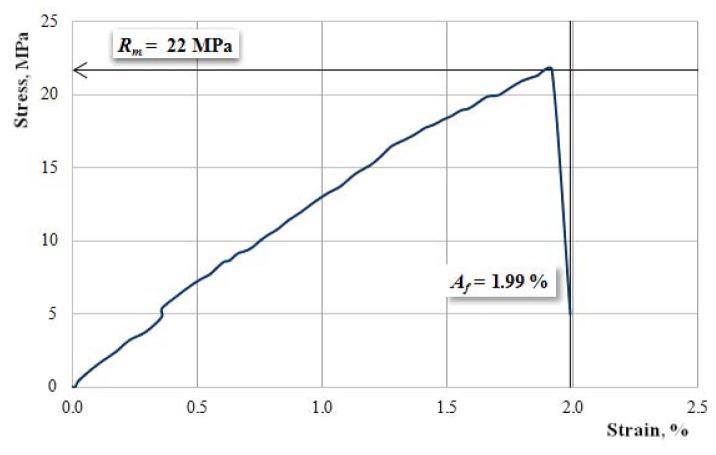
The characteristic curve of a Dammar-based resin sample.

**Figure 5 materials-12-01232-f005:**
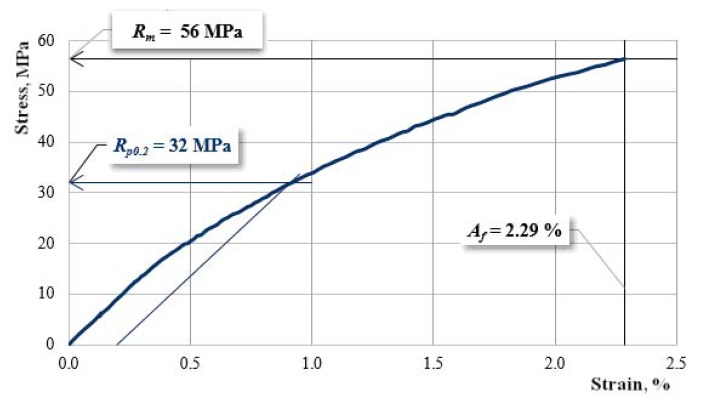
The characteristic curve for a sample A00.

**Figure 6 materials-12-01232-f006:**
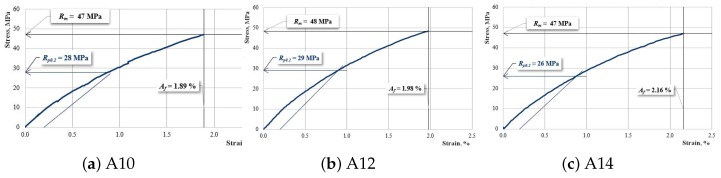
The characteristic curve for a sample: (**a**) A10; (**b**) A12; (**c**) A14.

**Figure 7 materials-12-01232-f007:**
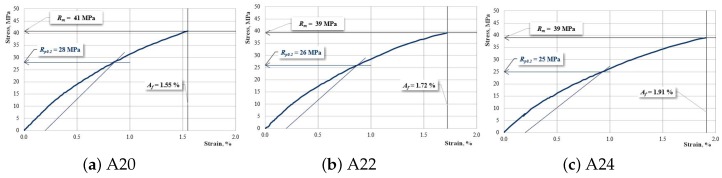
The characteristic curve for a sample: (**a**) A20; (**b**) A22; (**c**) A24.

**Figure 8 materials-12-01232-f008:**
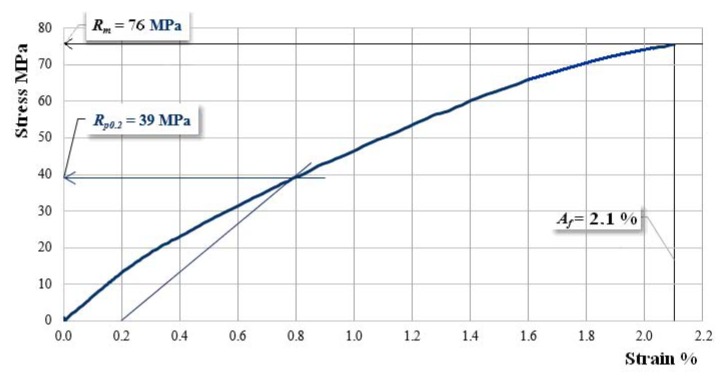
The characteristic curve for a sample B00.

**Figure 9 materials-12-01232-f009:**
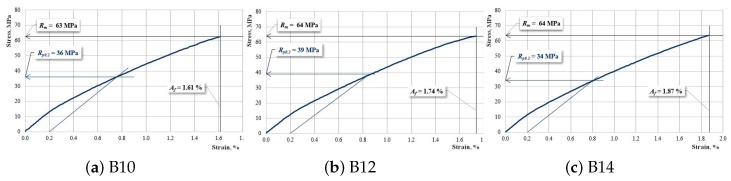
The characteristic curve for a sample: (**a**) B10; (**b**) B12; (**c**) B14.

**Figure 10 materials-12-01232-f010:**
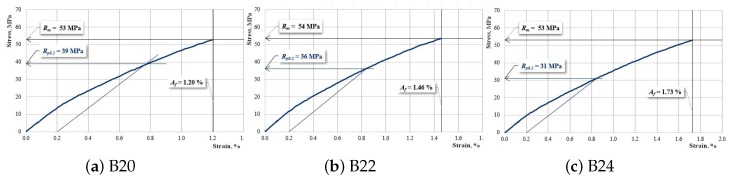
The characteristic curve for a sample: (**a**) B20; (**b**) B22; (**c**) B24.

**Table 1 materials-12-01232-t001:** Details about the manufactured samples.

Abbreviation Sample	The Total Number of Layers of Reinforcement	Number of Layers Interrupted	Length of Interruption [mm]	Number of Samples Tested
A00	5	0	0	10
A10	5	1	0	10
A12	5	1	20	10
A14	5	1	40	10
A20	5	2	0	10
A22	5	2	20	10
A24	5	2	40	10
B00	5	0	0	10
B10	5	1	0	10
B12	5	1	20	10
B14	5	1	40	10
B20	5	2	0	10
B22	5	2	20	10
B24	5	2	40	10

**Table 2 materials-12-01232-t002:** Type “A” hemp fabric reinforced samples: lower and upper values (arithmetical average value and deviation value) for the elasticity modulus, tensile strength, and elongation at break.

Abbreviation Sample	Modulus of Elasticity *E* [MPa]	Tensile Strength $R_{m}$ [MPa]	Elongation at Break *A* [%]
A00	4473–4622 (4547 ± 75)	55–57 (56 ± 1)	2.24–2.29 (2.27 ± 0.03)
A10	4440–4618 (4529 ± 89)	47–48 (47.5 ± 0.5)	1.89–1.95 (1.92 ± 0.03)
A12	4326–4445 (4386 ± 60)	47–48 (47.5 ± 0.5)	1.96–2.02 (1.99 ± 0.03)
A14	4090–4209 (4150 ± 60)	46–48 (47 ± 1)	2.08–2.16 (2.12 ± 0.04)
A20	4421–4573 (4497 ± 76)	40–41 (40.5 ± 0.5)	1.56–1.62 (1.59 ± 0.03)
A22	4062–4199 (4131 ± 68)	39–41 (40 ± 1)	1.71–1.77 (1.74 ± 0.03)
A24	3692–3790 (3741 ± 49)	38–40 (39 ± 1)	1.87–1.93 (1.90 ± 0.03)

**Table 3 materials-12-01232-t003:** Type “B” hemp fabric reinforced samples: lower and upper values (arithmetical average value and deviation value) for the elasticity modulus, tensile strength, and elongation at break.

Abbreviation Sample	Modulus of Elasticity *E* [MPa]	Tensile Strength $R_{m}$ [MPa]	Elongation at Break *A* [%]
B00	6580–6654 (6617 ± 37)	74–76 (75 ± 1)	2.02–2.10 (2.06 ± 0.04)
B10	6528–6610 (6569 ± 41)	62–64 (63 ± 1)	1.60–1.66 (1.63 ± 0.03)
B12	6094–6226 (6160 ± 66)	62–64 (63 ± 1)	1.70–1.74 (1.72 ± 0.02)
B14	5872–5989 (5930 ± 58)	61–64 (62.5 ± 1.5)	1.86–1.92 (1.89 ± 0.03)
B20	6568–6640 (6604 ± 36)	51–53 (52 ± 1)	1.21–1.29 (1.25 ± 0.04)
B22	5770–5899 (5834 ± 64)	50–54 (52 ± 2)	1.44–1.48 (1.46 ± 0.02)
B24	5257–5363 (5310 ± 53)	50–53 (51.5 ± 1.5)	1.72–1.82 (1.77 ± 0.05)

**Table 4 materials-12-01232-t004:** Quality factors for composites reinforced by type “A” fabric.

Sample Type	Elasticity Factor		Resistance Factor		Uniformity Factor	
	$f_{E}$		$f_{\sigma }$		$f_{u}$	
	Theoretical	Experimental	Theoretical	Experimental	Theoretical	Experimental
A10	1	0.991	0.853	0.848	0.853	0.855
A12	0.967	0.959	0.853	0.848	0.883	0.884
A14	0.936	0.908	0.853	0.839	0.912	0.924
A20	1	0.984	0.707	0.723	0.707	0.735
A22	0.923	0.903	0.707	0.714	0.765	0.790
A24	0.858	0.818	0.707	0.696	0.824	0.851

**Table 5 materials-12-01232-t005:** Quality factors for composites reinforced by type “B” fabric.

Sample Type	Elasticity Factor		Resistance Factor		Uniformity Factor	
	$f_{E}$		$f_{\sigma }$		$f_{u}$	
	Theoretical	Experimental	Theoretical	Experimental	Theoretical	Experimental
B10	1	0.993	0.836	0.840	0.836	0.846
B12	0.962	0.931	0.836	0.840	0.869	0.902
B14	0.927	0.896	0.836	0.833	0.902	0.929
B20	1	0.998	0.673	0.693	0.673	0.694
B22	0.911	0.882	0.673	0.693	0.738	0.785
B24	0.837	0.802	0.673	0.687	0.804	0.856

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
