# Peer review of "The Influence of Non-Uniformities on the Mechanical Behavior of Hemp-Reinforced Composite Materials with a Dammar Matrix"

_materials, 2019, doi:10.3390/ma12081232_

Round 1

Reviewer 1 Report

Dear Authors,

I have read the manuscript. Enlisted please find my comments.

Overall. Interesting manuscript about Hemp-Reinforced Composite Materials.

Abstract. Authors stated " we have studied the influence of the non uniformities on the mechanical behavior of the composites". It is too general in my opinion. Please reveal in advance some of the findings.

Introduction. Authors stated "The physical and chemical properties of composite materials depend on a lot of factors". Please add a reference for this statement.

Introduction. Authors stated "the mechanical behavior of composite materials is influenced by environmental factors (humidity, temperature, radiations, chemical agents), the mechanical stresses to which they are subjected (the stress type, the stress variation in time, the loading speed, the stress direction, the stress duration)". Please add reference for both environmental and mechanical factors.

Introduction. Authors stated that " The resin transfer, the structural reactions, the interface effects are phenomena that can be taken into account when considering the non uniformity degree". Please add a reference.

Introduction. Authors introduced fiber reinforced materials. In my opinion Authors could stress more the advantages of fiber reinforced composites over unreinforced materials and their application. They could specify that "Fiber-reinforced composites are composite materials with three different components: the matrix, the fibers, and the zone in between (interphase). Fiber reinforced composites are widely used for their advantages over unreinforced materials. Their applications include construction ( Short-term bond behavior and debonding capacity of prestressed CFRP composites to steel substrate. Hosseini, A., Ghafoori, E., Wellauer, M., Sadeghi Marzaleh, A., Motavalli, M. Engineering Structures Volume 176, 1 December 2018, Pages 935-947), aerospace ( Soutis, C. Fibre reinforced composites in aircraft construction. Prog. Aerosp. Sci. 2015, 41, 143–151. ), medicine (Finite element simulation of an artificial intervertebral disk using fiber reinforced laminated composite model.Shahmohammadi M, Asgharzadeh Shirazi H, Karimi A, Navidbakhsh M.Tissue Cell. 2014 Oct;46(5):299-303.) and dentistry (Fiber-Reinforced Composites for Dental Applications. Scribante A, Vallittu PK, Özcan M. Biomed Res Int. 2018 Nov 1;2018:4734986.) fields”. This paragraph could be added in Introduction section.

Introduction. Authors pointed out that The use of hemp fibres as reinforcement for composite materials has risen in the past few years. Before introducing hemp reinforced materials, it could be stated that ”Conventional Fiber reinforced composites present a reinforcement based on carbon, polyaramid, polyethylene, and glass) and can be used in a resin matrix (A retrofit theory to prevent fatigue crack initiation in aging riveted bridges using carbon fiber-reinforced polymer materials. Ghafoori, E, Motavalli, M, Polymers, Volume 8, Issue 8, 18 August 2016) or in combination with nanofillers (Effects of nanofillers on mechanical properties of fiber-reinforced composites polymerized with light-curing and additional postcuring. Scribante A, Massironi S, Pieraccini G, Vallittu P, Lassila L, Sfondrini MF, Gandini P. J Appl Biomater Funct Mater. 2015 Oct 16;13(3):e296-9). Fabricating composites by using natural fibers has been of great interest because natural fiber is biodegradable and environment friendly ( Mechanical, Degradation and Water Uptake Properties of Fabric Reinforced Polypropylene Based Composites: Effect of Alkali on Composites. Mohammad Bellal Hoque , Solaiman , A.B.M. Hafizul Alam , Hasan Mahmud  and Asiqun Nobi. Fibers 2018, 6(4), 94; )".

Experimental determinations. Authors  used the testing machine for mechanical trials LLOYD181 Instruments Lrx PLU. Please add Manufacturer, City and State.

Experimental determinations. Authors used the analysis software: NEXYGEN. Please point out Manufacturer, City and State.

Experimental determinations. Tables 1 and 2. The variables tested (Modulus of Elasticity, tensile strength and elongation at break) are reported as a range between two values. Please specify if the 2 values are minimum and maximum or other. Moreover please point out in the text if a statistical analysis has been performed. If so, please point out the names of the tests and the Software used.

Discussion. Authors stated “We believe that the reference values of the elastic modulus and tensile strength of the materials considered as ideal are the average values of the limits between which these sizes vary in samples  having all their layers without discontinuities”. Please ad a reference for these values.

References. In the text some sentences need a reference. Some recent references have been also suggested above.

Figures: ok.

Author Response

Dear Reviewer 1

In accordance with your last remarks, we have made the following changes in the paper structure:

Title: “THE INFLUENCE OF NONUNIFORMITIES ON THE MECHANICAL BEHAVIOR OF HEMP-REINFORCED COMPOSITE MATERIALS WITH A DAMMAR MATRIX

Authors: Dumitru Bolcu and Marius Marinel Stănescu

Abstract. Authors stated "we have studied the influence of the non uniformities on the mechanical behavior of the composites". It is too general in my opinion. Please reveal in advance some of the findings.

We specified what we mean by material irregularities (non uniformities). The results obtained are presented at the end of the abstract.

Introduction. Authors stated "The physical and chemical properties of composite materials depend on a lot of factors". Please add a reference for this statement.

At the request of another reviewer, we removed the paragraph that refers to the influence of some factors on the properties of composite materials.

Introduction. Authors stated "the mechanical behavior of composite materials is influenced by environmental factors (humidity, temperature, radiations, chemical agents), the mechanical stresses to which they are subjected (the stress type, the stress variation in time, the loading speed, the stress direction, the stress duration)". Please add reference for both environmental and mechanical factors.

We introduced the required references.

Introduction. Authors stated that "The resin transfer, the structural reactions, the interface effects are phenomena that can be taken into account when considering the non uniformity degree". Please add a reference.

We introduced the required references.

Introduction. Authors introduced fiber reinforced materials. In my opinion Authors could stress more the advantages of fiber reinforced composites over unreinforced materials and their application. They could specify that "Fiber-reinforced composites are composite materials with three different components: the matrix, the fibers, and the zone in between (interphase). Fiber reinforced composites are widely used for their advantages over unreinforced materials. Their applications include construction ( Short-term bond behavior and debonding capacity of prestressed CFRP composites to steel substrate. Hosseini, A., Ghafoori, E., Wellauer, M., Sadeghi Marzaleh, A., Motavalli, M. Engineering Structures Volume 176, 1 December 2018, Pages 935-947), aerospace ( Soutis, C. Fibre reinforced composites in aircraft construction. Prog. Aerosp. Sci. 2015, 41, 143–151. ), medicine (Finite element simulation of an artificial intervertebral disk using fiber reinforced laminated composite model. Shahmohammadi M, Asgharzadeh Shirazi H, Karimi A, Navidbakhsh M.Tissue Cell. 2014 Oct;46(5):299-303.) and dentistry (Fiber-Reinforced Composites for Dental Applications. Scribante A, Vallittu PK, Özcan M. Biomed Res Int. 2018 Nov 1;2018:4734986.) fields”. This paragraph could be added in Introduction section.

We introduced the required references.

Introduction. Authors pointed out that The use of hemp fibres as reinforcement for composite materials has risen in the past few years. Before introducing hemp reinforced materials, it could be stated that ”Conventional Fiber reinforced composites present a reinforcement based on carbon, polyaramid, polyethylene, and glass) and can be used in a resin matrix (A retrofit theory to prevent fatigue crack initiation in aging riveted bridges using carbon fiber-reinforced polymer materials. Ghafoori, E, Motavalli, M, Polymers, Volume 8, Issue 8, 18 August 2016) or in combination with nanofillers (Effects of nanofillers on mechanical properties of fiber-reinforced composites polymerized with light-curing and additional postcuring. Scribante A, Massironi S, Pieraccini G, Vallittu P, Lassila L, Sfondrini MF, Gandini P. J Appl Biomater Funct Mater. 2015 Oct 16;13(3):e296-9). Fabricating composites by using natural fibers has been of great interest because natural fiber is biodegradable and environment friendly (Mechanical, Degradation and Water Uptake Properties of Fabric Reinforced Polypropylene Based Composites: Effect of Alkali on Composites. Mohammad Bellal Hoque, Solaiman , A.B.M. Hafizul Alam , Hasan Mahmud  and Asiqun Nobi. Fibers 2018, 6(4), 94; )".

We introduced the required references.

Experimental determinations. Authors  used the testing machine for mechanical trials LLOYD181 Instruments Lrx PLU. Please add Manufacturer, City and State.

Experimental determinations. Authors used the analysis software: NEXYGEN. Please point out Manufacturer, City and State.

Experimental determinations. Tables 1 and 2. The variables tested (Modulus of Elasticity, tensile strength and elongation at break) are reported as a range between two values. Please specify if the 2 values are minimum and maximum or other. Moreover please point out in the text if a statistical analysis has been performed. If so, please point out the names of the tests and the Software used.

For each and every sample reinforced with "A" and "B" type hemp fabric, we give in Tables 2 and 3, the downer and upper values (arithmetical average value and deviation value) for the elasticity modulus, tensile strength and elongation at break. We have’t made a statistical analysis for those values simply because all the outcomes are strictly related to samples in study. The mechanical properties of hemp fibers are under many influences, so a statitical analysis in this case would have not be relevant.

Discussion. Authors stated “We believe that the reference values of the elastic modulus and tensile strength of the materials considered as ideal are the average values of the limits between which these sizes vary in samples  having all their layers without discontinuities”. Please ad a reference for these values.

The reference values are the arithmetical average values shown in Tables 2 and 3.

References. In the text some sentences need a reference. Some recent references have been also suggested above.

We introduced the required references.

Figures: ok.

We have marked the changes:

-             with green-color the answers for reviewer 1;

-             with yellow-color the answers for reviewer 2;

-             with red-color the answers for reviewer 3;

-             with blue-color the common answers for reviewers 1 and 3;

-             with brown-color the common answers for reviewers 2 and 3.

We thank you for your viewpoints based on which we have made the necessary changes to increase the scientific level of the article.

                                                                                             The authors

Reviewer 2 Report

An interesting paper that has scope for improvement.

Although the word appears in the title, there is insufficient information about Dammar (a resinous gum obtained from the Dipterocarpaceae family of trees in India and East Asia?)  What are the characteristics of this polymer?  Is this resin a commercial product or harvested from a tree in the laboratory garden?

Reference the review papers on hemp fibres: https://doi.org/10.1177/0021998311413623,

and on surface treatments applied to hemp: http://doi.org/10.1016/j.indcrop.2017.07.027 

Line 26: "many", not "a lot of"

Line 20: "spatial", not "space"

Line 29: "[r]einforced"

Line 30: "matrix fibres"?

Line 52: "fissure speed"?

Line 73: "viscosity", not "tack"?

Line 77: resins derived from a biological source are not always biodegradable or compositable!

Line 78: "they", not "the"

Line 90: the elastomer is "silicone" (silicon is a semiconductor element)!

Line 130 width b

Line 182: load cell and calibration thereof?

Line 218: "modulus", not "modules"

Author Response

Dear Reviewer 2,

In accordance with your last remarks, we have made the following changes in the paper structure:

Title: “THE INFLUENCE OF NONUNIFORMITIES ON THE MECHANICAL BEHAVIOR OF HEMP-REINFORCED COMPOSITE MATERIALS WITH A DAMMAR MATRIX

Authors: Dumitru Bolcu and Marius Marinel Stănescu

Although the word appears in the title, there is insufficient information about Dammar (a resinous gum obtained from the Dipterocarpaceae family of trees inIndiaand East Asia?)  What are the characteristics of this polymer?  Is this resin a commercial product or harvested from a tree in the laboratory garden?

We gave some data about Dammar resin.

Reference the review papers on hemp fibres: https://doi.org/10.1177/0021998311413623,

and on surface treatments applied to hemp: http://doi.org/10.1016/j.indcrop.2017.07.027 

We made required changes, and bibliographic references.

Line 16: "many", not "a lot of"

Line 20: "spatial", not "space"

Line 29: "[r]einforced"

Line 30: "matrix fibres"?

Line 52: "fissure speed"?

Slowing crack propagation fissure speed

Line 73: "viscosity", not "tack"?

Line 77: resins derived from a biological source are not always biodegradable or compositable!

We made the change.

Line 78: "they", not "the"

Line 90: the elastomer is "silicone" (silicon is a semiconductor element)!

Line 130 width b

Line 182: load cell and calibration there of?

This data can be found in the characteristics of testing machine (manufacturer data are cited in the references).

Line 218: "modulus", not "modules"

We have marked the changes:

-             with green-color the answers for reviewer 1;

-             with yellow-color the answers for reviewer 2;

-             with red-color the answers for reviewer 3;

-             with blue-color the common answers for reviewers 1 and 3;

-             with brown-color the common answers for reviewers 2 and 3.

We thank you for your viewpoints based on which we have made the necessary changes to increase the scientific level of the article.

                                                                              The authors

Reviewer 3 Report

Dear Authors, 

thank you for this manuscript. In my opinion it is fully centered, both in terms of Journal and Special Issue. At the same time, it is quite pour in terms of novelty and significance. Half part of the introduction is too general and potentially useless (for a reader of MATERIALS). The other half part is almost OK, but not appropriate to fully define the general context your research is moving thru'. For instance, it seems to me that several references to similar researches are missing. 

Beyond this fact, almost marginal respect to others, the real problem is in the research implementation. Experiments are quite limited to few tests, without a clear indication of a 'design for experiments'. 14 measures are reported and analyzed, maybe too few for finding consolidated results and to propose general considerations. Adding, data are not correctly considered, especially in terms of variability, experimental errors and so on. As scientific elements to be appreciated, the part regarding the discussion is very good, even if a bit oriented a-priori by the conclusions. The introduction of dditional tables and graphs could be really useful for a better explaination of concepts.

My general suggestion is related to a preliminary extension/repeatition of the experimental tests before submitting the article again. Several specific suggestions are available in the attachment. English has to be improve.

Author Response

Dear Reviewer 3

In accordance with your last remarks, we have made the following changes in the paper structure:

Title: “THE INFLUENCE OF NONUNIFORMITIES ON THE MECHANICAL BEHAVIOR OF HEMP-REINFORCED COMPOSITE MATERIALS WITH A DAMMAR MATRIX

Authors: Dumitru Bolcu and Marius Marinel Stănescu

Professor Bolcu has no ID.

We say what we mean by non uniformities.

Are you sure about the correctness of this term to describe your manufacturing process ?

We changed "cast" with "molded".

elasticity is a mechanical property too.

We removed elasticity.

which curves

By the characteristic curve we understood the stress-strain curve.

you did not say anything in the abstract about hybrid resins. And this word (hybrid) is not even used in the introduction. So it is a bit undefined.

We introduced "hybrid resin" in the abstract.

In introduction we specified what is hybrid resin.

Line 16-30 are almost useless in this form. The report general concepts everyone knows, even at the level of students. You have to delete or focus them in some way.

We removed these lines.

also in this case, line 54-64 are almost useless in this form. Adding, references 11-15 are included without any definition or specific description. In this way, it is possible to add hundreds of papers.

We removed some of the specified lines and introduced new bibliographic references with specific description.

which properties are you interested in particular ? which results can be included from these articles inside your introduction? ... and so on..

We are interested in mechanical properties that characterize the mechanical behavior of traction, torsion and bending for composite materials with polypropylene matrix and hemp reinforcement.

in the keywords, you refereed to hybrid resin. Nothing in the introduction explicitly refers to that. It is better to add a sentence referring to what you intend for hybrid resin and to add some references.

We corrected this.

furthermore, it could be usefull do add some words about an environmental alternative to these hybrid matrix, offered by 'ecoresins' (low emission resins).

Some practical & useful notes about the problem can be found in the conference papers (not easy to be found):

- De Paola S, Minak G, Fragassa C, Pavlovic A (2013) Green Composites: A Review of State of Art. In: Proc. 30th Danubia Adria Symposium on Advanced Mechanics. Croatian Society of Mechanics (ed). Primosten,Croatia, 25-28 September 2013 [ISBN: 978-953-7539-17-7]: pp 77-78.

- Hyseni A, De Paola S, Minak G, Fragassa C (2013) Mechanical Characterization of EcoComposites. In: Proc. 30th Danubia Adria Symposium on Advanced Mechanics. Croatian Society of Mechanics (ed), Primosten,Croatia, 25-28 September 2013 [ISBN: 978-953-7539-17-7]: pp 175-176.

Larger investigations about hybridisation (in fibers and/or matrices), are available in:

- Zivkovic I, Pavlovic A, Fragassa C, Brugo T (2017). Influence of moisture absorption on the impact properties of flax, basalt and hybrid flax/ basalt fiber reinforced green composites. Composites Part B, Vol. 111: pp 148-164. Doi 10.1016/j.compositesb.2016.12.018

- Fragassa C, Pavlovic A, Santulli C (2018). Mechanical and impact characterisation of flax and basalt fibre bio-vinylester composites and their hybrids. Composites - Part B. Vol 137, pp. 247-259. doi: 10.1016/j.compositesb.2017.01.004

We referred to the required bibliographic references.

it is unclear the overall experimental design. Also considering that some of these specimens are not reported ib terms of results. It is better to report a table able to summarize an approach toward the 'design for experiment'. In particular, number of sets, of samples, different condition of tests and so on.

We gave a table in which we presented details of the test samples.

adding, according to this first sight, you simply produced two different sets/conditions (in terms of materials). Then, pay attention about the general conclusion you are going to extrapolate from this one-to-one comparison.

We tested to tensile test a number of 14 sets of samples, each set with 10 samples (without resin samples).

please, consider that scientific articles used to be written in an impersonal form

We fixed the writing to the impersonal form.

each formula has to be refer to the theory and to the literature.

We inserted two new bibliographical references.

number missing

We attached numbers to all equalities you indicated us.

if possible, modify this graph in the way to have the same y-axis range [0-60] for an easier comparison with the others.

Do the same for graphs in the following picture.

We changed the scale of the indicated graphs.

what is the reason why you split these graph in two pictures ? it makes sense if graphs are related to different groups (but I do not think it is the case)

Figure 5 present the characteristic curve for a sample reinforced with hemp fabric of "A" type, without reinforcement interrupted layers.

Figure 6 present the characteristic curves for samples reinforced with hemp fabric of "A" type, which have an interrupted layer of reinforcement.

Figure 7 present the characteristic curves for samples reinforced with hemp fabric of "A" type, which have two interrupted layers of reinforcement.

Figure 8 present the characteristic curve for a sample reinforced with hemp fabric of "B" type, without reinforcement interrupted layers.

Figure 9 present the characteristic curves for samples reinforced with hemp fabric of "B" type, which have an interrupted layer of reinforcement.

Figure 10 present the characteristic curves for samples reinforced with hemp fabric of "B" type, which have two interrupted layers of reinforcement.

how can you obtain a range ?

add average values and (standard) variation in each coloumn

For each and every sample reinforced with "A" type hemp fabric, we give below (Table 1) the downer and upper values (arithmetical average value and deviation value) for the elasticity modulus, tensile strength and elongation at break.

For each and every sample reinforced with "B" type hemp fabric, we give below (Table 2) the downer and upper values (arithmetical average value and deviation value) for the elasticity modulus, tensile strength and elongation at break.

I suggest to split the Conclusion from the Discussion in the way to clarify their related role.

We split the discussions and conclusions section.

these results surely come from the logics, but it is not fully clear how they can be derived from the data. At this stage, you simply reported 2 table with measures on 2x7 samples. Maybe you can say something like that at the end of the further considerations.

This part has to be widely rearranged and, maybe, extended.

We added some elements that complement the conclusions.

it is better to add a nomenclature at the end of the article

We added a nomenclature at the end of the paper.

to support these general considerations and conclusions, maybe could be convenient to add graphs compating specific results.

Based on the existing graphs, we completed the conclusions.

We have marked the changes:

-             with green-color the answers for reviewer 1;

-             with yellow-color the answers for reviewer 2;

-             with red-color the answers for reviewer 3;

-             with blue-color the common answers for reviewers 1 and 3;

-             with brown-color the common answers for reviewers 2 and 3.

We thank you for your viewpoints based on which we have made the necessary changes to increase the scientific level of the article.

                                                                              The authors

Round 2

Reviewer 1 Report

Good job.

All concers have been solved.

Reviewer 3 Report

No additional comments.

Materials EISSN 1996-1944 Published by MDPI AG, Basel, Switzerland RSS E-Mail Table of Contents Alert
Back to Top